



# Increasing boreal wetland emissions inferred from reductions in atmospheric CH$_4$ seasonal cycle

J. M. Barlow[1], P. I. Palmer[1], and L. M. Bruhwiler[2]

[1]School of GeoSciences, University of Edinburgh, UK
[2]National Oceanic and Atmospheric Administration, Earth System Research Laboratory, Boulder, Colorado, USA.

*Correspondence to:* P. I. Palmer
(paul.palmer@ed.ac.uk)

**Abstract.** Observed variations of the atmospheric greenhouse gas methane (CH$_4$) over the past two decades remain the subject of debate. These variations reflect changes in emission, uptake, and atmospheric chemistry and transport. We isolate changes in the seasonal cycle of atmospheric CH$_4$ using a wavelet transform. We report a previously undocumented persistent decrease in the peak-to-

peak amplitude of the seasonal cycle of atmospheric CH$_4$ at six out of seven high northern latitude sites over the past two to three decades. The observed amplitude changes are statistically significant for sites at Barrow, Alaska and Ocean Station M, Norway, which we find are the most sensitive of our sites to high northern latitude wetland emissions. We find using a series of numerical experiments using the TM5 atmospheric chemistry transport model that increasing wetland emissions and/or

decreasing fossil fuel emissions can explain these observed changes, but no significant role for trends in meteorology and tropical wetlands. We also find no evidence in past studies to support a significant role for variations in the hydroxyl radical sink of atmospheric CH$_4$. Using the TM5 model we find that changes in fossil fuel emissions of CH$_4$, described by a conservative state-of-the-science bottom-up emission inventory, are not sufficient to reconcile observed changes in atmospheric CH$_4$ at these

sites. The remainder of the observed trend in amplitude, by process of elimination, must be due to an increase in high northern latitude wetland emissions, corresponding to an annual increase of at least 0.7%/yr (equivalent to 5 Tg CH$_4$/yr over 30 years).

## 1 Introduction

Atmospheric CH$_4$ is a potent greenhouse gas with a global budget that is determined by natural and

anthropogenic sources, and its atmospheric loss due to reaction with hydroxyl (OH) radicals. The resulting atmospheric lifetime is approximately nine years. Changes in atmospheric CH$_4$ over the past two decades have included a slowing of the growth rate in the late 1990s, reaching an approximate steady state from 1999 until 2007 (Dlugokencky et al., 1998, 2003, 2009) after which it resumed its growth rate in the atmosphere (Rigby et al., 2008; Dlugokencky et al., 2009; Nisbet et al., 2014;



Zona et al., 2016). Our understanding of the underlying reasons for these changes is incomplete (e.g., Bloom et al. (2010); Kai et al. (2011); Simpson et al. (2012); Schaefer et al. (2016)). The continued growth of $CH_4$ in the global atmosphere since 2007 has been linked to a response of high northern latitude wetlands to anomalously warm temperatures during 2007, and increased precipitation over tropical wetlands during 2008-2011, a period of increased La Ninã activity (Dlugokencky et al., 2009). To draw these conclusions previous studies have tended to focus on year-to-year changes of the annual global growth rate of $CH_4$ and/or concurrent temporal variations in atmospheric $CH_4$ isotopologues. Here, we show that observed changes in the peak-to-peak amplitude of the seasonal cycle of atmospheric $CH_4$, isolated using a wavelet transform (Barlow et al., 2015) provide additional information about changes in high northern latitude fluxes of $CH_4$.

The next section describes the data and methods we use to interpret observed variations of atmospheric $CH_4$ mole fraction. In section 3 we describe our results. We conclude our study in section 4. To improve the readability of the report we have placed many of the details that describe the underpinning data analysis and mathematical modelling in Appendices.

## 2   Data and Methods

Figure 1 shows the geographical locations of the $CH_4$ mole fraction time series we analyze. For this study we use five types of data (described in detail in Appendix A): 1) hourly averaged continuous $CH_4$ analyzed using a gas chromatograph; 2) discrete flask $CH_4$ measurements collected on at least a weekly basis and sampled for off-shore wind conditions to minimize the influence of local sources (Dlugokencky et al., 2005); 3) coincident $\delta^{13}CH_4$ isotope measurements analysed from the $CH_4$ flasks (White and Vaughn, 2011); 4) local meteorology including wind speed and direction and two-metre temperatures; and 5) Modern Era Retrospective-Analysis for Research (MERRA) gridded temperature time series (Rienecker et al., 2011). We report results based on our analysis of the flask data, which were imputed and averaged on a monthly timescale. Analysis of weekly averaged continuous data that are sampled for offshore wind conditions independently verify our results.

We use a wavelet transform, described in detail in Appendix B (Torrence and Compo, 1998; Barlow et al., 2015; Mackie et al., 2016), to spectrally decompose the flask $CH_4$ mole fraction data collected at seven high northern latitude sites (Figure 1). The wavelet transform decomposes a time series into time–frequency space, allowing us to investigate the dominant modes of variability and how they change with time. This improves on the Fourier transform that determines frequency information using sine and cosine functions. We focus on high northern latitude sites because they are expected to be most sensitive to high-latitude wetland emissions. For the purposes of our analysis, guided by the power spectrum of these data (not shown), we retain periods between 2–18 months that describe sub-annual and annual frequencies and define this as the seasonal cycle. Periods greater than 18 months represent the long-term changes in $CH_4$ from which the inter-annual growth rate





can be calculated. We discard frequencies less than two months, which represent fast, small-scale variations.

To understand how increases in fossil fuel and wetland emissions and inter-annual meteorological variations in meteorology manifest themselves in the $CH_4$ seasonal cycle at high northern latitudes we use the TM5 atmospheric chemistry transport model (Krol et al., 2005). We use prior emission

estimates for natural and anthropogenic sources and for the soil sink, which are described in detail in Appendix C. We use a prescribed repeating annual cycle of monthly mean 3-D fields of the OH sink, allowing us to linearize the chemistry so that we can attribute observed variations to individual sources and geographical regions using tagged tracers. We use tracers for emissions from fossil fuel combustion, agriculture, and natural wetlands. The TM5 model in previous studies has shown skill

at reproducing observed variations of $CH_4$ at the background atmospheric network sites (Bruhwiler et al., 2014).

We run 12 numerical experiments from 1980 to 2010: E1) A repeating annual cycle of wetland emissions and time-dependent meteorology (control); E2) as the control, but using a repeated year of meteorology; E3)-E5) as the control, but using a progressive increase in Arctic wetland emissions

(0.5 %/yr, 1 %/yr, and 2 %/yr); and E6) as the control but including a progressive increase in tropical wetland emissions from 2007 onwards. We run each of these simulations twice, once with a repeated year of anthropogenic emissions (EDGAR emissions for 2002, Appendix C), and once with time-dependent EDGAR anthropogenic emissions.

The control simulation allows us to determine the contribution to interannual variability in sea-

sonal amplitude produced by variations in atmospheric transport alone and also acts as a reference time series with which to compare the results from the different emission scenarios. The emission scenarios are the same as the control simulation but with incremental increases in high northern latitude wetland $CH_4$ emissions of 0.5, 1 and 2%/yr respectively relative to an initial value of 25 Tg/yr, where the emissions increase is symmetric across the growing season. The high-latitude region is de-

fined as an aggregation of the Transcom 3 regions boreal North America, boreal Eurasia and Europe. As some of Europe lies to the south of what might typically be considered high northern latitudes, the area of Europe north of 47°N is used in the definition of the high-latitude region as this roughly corresponds with the southern extent of boreal North America and Eurasia (Bruhwiler et al., 2014). To compare model output with $CH_4$ mole fraction data, the 3D field of model $CH_4$ concentrations is

sampled at the surface level at the nearest grid point to each of the NOAA/ESRL monitoring sites. The resulting time series is made up of 365 data points each year, which are then averaged weekly for consistent comparison with the $CH_4$ mole fraction data.

The sampled time series are decomposed using the wavelet transform into sub-annual, annual and trend components using the criteria defined above. First, we check for consistency between the model

and data by comparing the mean seasonal cycles from the model output and $CH_4$ data respectively.





We then use the model time series for analysis of the seasonal amplitude, as described for the $CH_4$ mole fraction data.

## 3 Results

**Interpretation of Observations**

We determine the amplitude of the seasonal cycle using the peak-to-peak difference in one seasonal cycle that typically straddles successive calendar years. We find that most sites show a decrease in amplitude over their record, but the two sites that exhibit statistically significant trends are BRW and STM (Figure 1, Table 1). Similar analysis using the discrete flask data support the results using the monthly data, but are more sensitive to imputation, as expected; similar analysis using the con-
tinuous BRW data also support the analysis of the flask data. Analysis of other sites are shown for completeness in Appendix D.

Possible factors that could be responsible for a persistent reduction in the seasonal cycle of $CH_4$ at high northern latitudes include: 1) a trend in the OH sink at these high latitudes, 2) a trend in $CH_4$ due to concentration variations in OH at lower latitudes that is subsequently transported to higher
northern latitudes; 3) a change in a seasonally-varying emission source; or 4) a trend in meteorology transporting air masses into the region. Observational evidence does not support persistent large-scale changes in the OH sink (Montzka et al., 2011; Patra et al., 2014; Mackie et al., 2016).

Two independent observational datasets are consistent with a potential larger source from wetlands: 1) isotopologues of atmospheric $CH_4$ and 2) trends in two metre temperatures.

Isotope samples are sparse but recent work has suggested that the renewed growth of atmospheric $CH_4$ is related to microbial processes (Schaefer et al., 2016). Figure 2 shows our analysis of the $\delta^{13}$C-$CH_4$ ratio at ALT and BRW. Each Keeling plot was calculated using at least five coinciding measurements in each window of nine measurements across the time series despite some years having sparse measurements. The majority of intersects $>$-60‰ tend to occur in spring and winter whereas the
opposite is true for summer and autumn. The time series mean intersects for Summer and Autumn are -68.79±13.55‰ and -69.47±13.55‰ for ALT and -68.10±12.90‰ and -64.74±4.81‰ for BRW respectively. Most isotopic wetland signatures tend to lie in this region, where for example the signatures associated with worldwide boreal wetland emissions are typically between -69 to -65‰ (Sriskantharajah et al., 2012).

Figure 3 shows that two-metre temperatures at BRW typically rise above and drop below 0°C in June and September, respectively. We find that this period above 0° has get been getting earlier by -3.0±2.6 days/decade (p<0.01) and and getting later by +5.6±3.1 days/decade (p<0.01) from 1976 to 2012. The result is a progressive widening of the time period over which the temperature is greater than 0°C, implying warmer soils and larger $CH_4$ production from methanogenesis. This is supported
on large geographical scales by the analysis of MERRA temperature time series, which indicates





significant spatial variation in the lengthening of the warm period, but with an average value across North American and Eurasian permafrost affected soils of 3.4±1.1 days/decade (p<0.01) (Figure 1).

Using methods described in Appendix D we isolate local $CH_4$ variations at BRW by subtracting monthly averaged onshore continuous measurements, identified by wind direction, from the offshore. Local temperature variations explain only 18% of the observed $CH_4$ variations but it is statistically significant with sensitivity of 0.47±0.42 ppb per additional day above > 0°C (p<0.05).

**Interpretation of Numerical Experiments**

We find that meteorological variations account for large inter-annual variations in the seasonal cy-
cle amplitude of $CH_4$ but do not result in a long-term trend (E1—E2). In general, the seasonal amplitude at high-latitude sites appears to be insensitive to the imposed increasing Arctic wetland emissions (E3—E5). The exception is at BRW where the seasonal amplitude progressively decreases by -0.39±0.38 ppb/yr for the 2%/year increase in wetland emissions (Appendix D), due to a less pronounced summertime minima, consistent with observed variations. An increase in tropical wetland
emissions from 2007 results in a small increase in the seasonal amplitude of $CH_4$ at high northern latitude, which is inconsistent with observed changes.

Time-dependent changes in fossil fuel emissions result in a decrease in the amplitude of the seasonal cycle (Figure 4), qualitatively consistent with observed changes in atmospheric $CH_4$ mole fractions, by virtue of a lower winter maxima. While most sites are affected by changes in fossil
fuel emissions (Dlugokencky et al., 2003) they are only significant at BRW. Changes in fossil fuel emissions lead to a decrease in amplitude at BRW of -0.45±42 ppb/yr which explains ≃75% of the observed trend. Consequently, a smaller coincident increase in high-latitude wetland emissions of 0.73%/yr (equivalent to 5.4Tg/yr over 30 years) is necessary to reproduce the observed trend (Figure 4). Reported biases for EDGAR (Bergamaschi et al., 2013; Bruhwiler et al., 2014) suggest that our
wetland emission trend could be an underestimate.

**4    Discussion and Concluding remarks**

We isolated the seasonal cycle of atmospheric $CH_4$ using a wavelet transform, a method we have characterized for this purpose. We reported a previously undocumented persistent decrease in the peak-to-peak amplitude of this seasonal cycle at six out of the seven Arctic sites over the past two
decades, but we find only observed variations at Barrow, Alaska and Ocean Station M, Norway are statistically significant. Using measurements and a global 3-D atmospheric transport model we discounted a significant role for trends in meteorology, tropical wetlands, and the hydroxyl radical sink of atmospheric $CH_4$. We hypothesized that increasing high northern latitude wetland emissions and/or decreasing fossil fuel emissions could explain this observation. Using the EDGAR bottom-



up emission inventory, which is known to have be a positive bias against atmospheric measurements of $CH_4$, we could not quantitatively reconcile observed and model trends in the amplitude of the seasonal cycle. Only by including an increase in high northern latitude wetlands of 0.73%/yr could the model quantitatively match the observed changes in atmospheric $CH_4$.

    The pan-Arctic significance of our results is unclear. We find that many high northern latitude

sites exhibit low sensitivity to changes in high latitude wetland sources. This is partly because the high northern latitude emissions and sinks are proportionally small when compared to those transported from the mid-latitudes that contribute a large part of the observed seasonal cycle. It is also due to the timing of the peak wetland contribution coinciding with the ascending edge of the observed $CH_4$ seasonal cycle, with minimum overlap with either the peak or trough of the seasonal cycle

(Appendix D). Consequently, small changes in the seasonal amplitude metric that we describe in this paper are difficult to detect with the current measurement network configuration. Analysis using the HYSPLIT Lagrangian atmospheric dispersion model (Appendix D) shows that the BRW $CH_4$ seasonal cycle is the most sensitive to changes in high northern latitude wetland emissions because of its proximity to extensive areas of wetlands. At ALT and ZEP we find significant but infrequent

contributions from Arctic wetlands, while BRW often sees significant wetland signals, consistent with our large-scale TM5 numerical experiments. Our numerical experiments, following other studies, assume symmetrical wetland emissions about the growing season, and thus earlier or delayed spring and autumn warming and enhanced methanogenesis at the edges of the growing season could make the amplitude metric more or less sensitive to wetland emissions, depending on the location of

the site.

    Current climate projections show strong regional warming of ecosystems, which will respond by increasing biological activity and subsequently releasing more $CH_4$ into the atmosphere. A focus on interpreting large-scale, annual changes in atmospheric $CH_4$ has diverted attention from observed changes in its seasonal cycle that are consistent with a small but persistent increase in wetland

emissions and could potentially signify the start of a positive climate feedback process. Disproving our hypothesis requires an expansion of the ground-based network at the high northern latitudes that takes into account the most vulnerable of the boreal ecosystems, but even after ignoring associated financial constraints this is non-trivial because of the logistical challenges of pan-Arctic monitoring.

*Acknowledgements.* We thank NOAA/ESRL for the $CH_4$ surface mole fraction data which is provided by

NOAA/ESRL PSD, Boulder, Colorado, USA, from their website http://www.esrl.noaa.gov/psd/. J.M.B., P.I.P. and L.M.B. designed the statistical tests and computation experiments and L.M.B. led the TM5 model experiments. P.I.P. wrote the manuscript. J.M.B. was funded by United Kingdom Natural Environmental Research Council studentship NE/1528818/1, and P.I.P. gratefully acknowledges his Royal Society Wolfson Research Merit Award.



## Appendix A: Data Used

**Continuous CH$_4$ Measurements and Meteorology at BRW**

To estimate CH$_4$ anomalies associated with local wetland emissions on the Alaska North Slope, we use hourly averaged continuous CH$_4$ measurements at Barrow, Alaska (BRW). CH$_4$ was measured by gas chromatography with flame ionization detection. All measurements used are on the WMO X2004 CH$_4$ mole fraction scale (Dlugokencky et al., 2005). The repeatability of the CH$_4$ measurements ranges from 1 to 3 ppb, representing $1\sigma$ of 20 measurements of a gas standard (Dlugokencky et al., 1995).

Measurements collected when the analytical instrument was not working properly are flagged using a rule-based editing algorithm (Masarie et al., 1991). Hourly averages are calculated from $2-3$ individual measurements per hour from $1986-1996$ and 4 measurements per hour from 1996 to present. Local meteorological data are used to separate measurements by air sector (defined by wind direction). Hourly averages of temperature measured at 2 m above ground are used for comparison with CH$_4$ emission estimates, where missing values in the hourly temperature time series are interpolated.

**Discrete CH$_4$ Measurements**

We use weekly values and monthly averages of CH$_4$ from measurements of discrete air samples collected in flasks at a number of high-latitude sites shown in Figure 1 from the NOAA Cooperative Global Air Sampling Network (NOAA CGASN). Air samples (flask) are collected at the sites and analysed for CH$_4$ at NOAA ESRL in Boulder, Colorado using a gas chromatograph with flame ionization detection.

Each sample aliquot is referenced to the WMO X2004 CH$_4$ standard scale (Dlugokencky et al., 2005). Individual measurement uncertainties are calculated based on analytical repeatability and the uncertainty in propagating the WMO CH$_4$ mole fraction standard scale. Analytical repeatability has varied between 0.8 to 2.3 ppb, but averaged over the measurement record is approximately 2 ppb. Uncertainty in scale propagation is based on a comparison of discrete flask-air and continuous measurements at the MLO and BRW observatories and has a fixed value 0.7 ppb. These two values are added in quadrature to estimate the total measurement uncertainty, equivalent to a $\sim$68% confidence interval.

These monitoring sites usually collect at least one air sample per week, justifying our analysis of weekly values or monthly averages. The wavelet decomposition method, described below, requires a continuous time series with a constant spacing so we impute any missing data points in the measurement time series. First, we subtract the long-term trend from a reference time series which is representative of the latitude band, in this case BRW. Second, we calculate a local time-averaged seasonal cycle and extract the missing value from this seasonal cycle before adding the trend value





from the reference time series. This step ensures that imputed data points are weighted by variability
in the actual data under the assumption that the atmospheric growth rate at any particular site is sim-
ilar to BRW. Finally, any remaining missing datapoints are extracted from a piecewise cubic spline
curve fit. Sections of the time series that contain significant portions of missing data will exhibit spu-
rious variations. However, we find that missing sections are unusually and that the isolated periods
of missing data are not long enough to significantly impact the determination of long-term trends.

    Figure 5, for example, shows the discrete $CH_4$ flask measurements at BRW and imputed data. The
first $5-6$ years of these data contain the largest proportion of missing data but generally there are
25% missing data points per year that tend to be spread sporadily within a particular year. Based on
this observation, the earliest years of the BRW time series are likely to be the most unreliable.

**$CH_4$ Isotope Record ($\delta^{13}$C-$CH_4$)**

    We also use weekly measurements of the stable isotopic composition ($^{13}$C) of atmospheric $CH_4$,
$\delta^{13}$C-$CH_4$ (White and Vaughn, 2011). The $\delta^{13}$C-$CH_4$ measurements at ALT and BRW spans $1990-2012$
and $1998-2012$, respectively, and are the only high-latitude records with a time span of $>10$ years.
The $\delta$ notation refers to the ratio of minor to major isotopes relative to a standard:

$$\delta^{13}C_{sample} = \left[ \frac{(^{13}C/^{12}C)_{sample}}{(^{13}C/^{12}C)_{std}} - 1 \right] \times 1000, \tag{A1}$$

and is expressed in units of permil (parts per thousand). The isotope samples are analysed at the
Stable Isotope Laboratory at CU-INSTAAR in Boulder, Colorado, where repeatability is estimated
to be approximately 0.1 permil for $^{13}$C.

    We use the Keeling plot approach (Pataki et al., 2003) to assess bulk inputs of $CH_4$ into Arc-
tic air. This approach involves using geometric mean regression to find the intercept of $\delta^{13}$C-$CH_4$
and 1/$CH_4$, a value which can be associated with the $CH_4$ source. We use a running window of
nine weeks across the 1/$CH_4$ and $\delta^{13}$C-$CH_4$ time series and produce a Keeling plot, calculating the
intercept for each window, assuming that the window has at least 5 coinciding measurements. The
geometric mean regression provides the intercept and 95% confidence interval for each Keeling plot.
We also calculate the mean intersect by season for each year. The three primary classes of $CH_4$ have
distinct isotopic signatures that have a characteristic range of values, $\delta^{13}$C≈-60‰ for microbial
$CH_4$, $\delta^{13}$C≈-40‰ for thermogenic $CH_4$ and $\delta^{13}$C≈-20‰ for biomass burning $CH_4$ (Quay et al.,
1991). While individual sources of $CH_4$ are likely to be significantly different from the characteris-
tic signature of an individual source, the average values are likely to be valid for large spatial scales
(Conny and Currie, 1996).

   **Ancillary Data**

    Trends and interannual variability in $CH_4$ are compared with gridded temperature (resolution of
$1°$ latitude $\times$ 2/3$°$ longitude) from the Global Modeling and Assimilation Office (GMAO) Mod-





ern Era-Retrospective Analysis for Research and Applications (MERRA) dataset (Rienecker et al.,
2011). MERRA covers the time span of modern era remotely sensed data (1979−present), and also
overlaps with the period of consistent ground-based observations of surface $CH_4$ concentrations
from NOAA CGASN. We sample gridded surface temperatures over areas of Eurasian and North
American permafrost affected wetlands (see Figure 1) and build up time series of growing season
temperature anomalies and potential growing season length (where the potential growing period is
defined as days$>0°$) for each region. To analyse temperature trends over high latitude wetland ar-
eas, we use the soil carbon maps provided in Northern Circumpolar Soil Carbon Database version 2
(NCSCDv2, (Hugelius et al., 2013)).

**Appendix B: Wavelet Transform**

We use a wavelet transform to spectrally decompose the observed $CH_4$ time series before recon-
structing time series with periods of 2-18 months to describe the seasonal cycle and $>18$ months to
describe long-term changes from which the annual growth rate can be calculated.

In general a wavelet transform $W_n$ uses a wavelet function $\psi_0$, a pre-defined wave-like oscillation
that is non-continuous in time or space, to decompose a time series into time-frequency space, allow-
ing us to investigate the dominant modes of variability and how they change with time. This improves
on the Fourier transform that determines frequency information using sine and cosine functions.

The wavelet transform of a time series $x_n$ is defined as

$$W_n(s) = \sum_{k=0}^{N-1} \hat{x}_k \hat{\psi} * (s\omega_k) e^{i\omega_k n \delta t} \tag{B1}$$

where $\hat{x}_k$ is the discrete Fourier transform of $x_n$, $N$ is the number of points in the time series,
$k=0...N-1$ is the frequency index and $\hat{\psi} * (s\omega_k)$ is the complex conjugate of the Fourier transform
of a normalized, scaled and translated version of $\psi_0(\eta)$, where $s$ is the scale and $\omega_k$ is the angular
frequency. We use the Morlet wavelet (Torrence and Compo, 1998), a plane wave modulated by a
gaussian envelope:

$$\psi_0(\eta) = \pi^{-1/4} e^{i\omega_0 \eta} e^{-\eta^2/2} \tag{B2}$$

where $\omega_0$ is the nondimensional frequency and $\eta$ is the non-dimensional time-parameter. We chose
the Morlet wavelet because it is nonorthogonal, which is an attractive property for the analysis of
smooth and continuous variations such as those exhibited by $CH_4$ mole fraction time series. The
wavelet is comprised of a real and imaginary part, providing information about amplitude and phase,
respectively.

We can recover the original time series from wavelet space using the corresponding inverse trans-
form (Torrence and Compo, 1998) and summing over all frequencies from the real part of the wavelet





transform (or a subset of frequencies if we are interested in isolating signals):

$$\hat{W}_n = \frac{\delta j \delta t^{1/2}}{C_\delta \psi_0(0)} \sum_{j=0}^{J} \frac{\Re\{W_n(s_j)\}}{s_j^{1/2}}, \tag{B3}$$

where $\psi_0(0)$ removes the energy scaling and $s_j^{1/2}$ converts the wavelet transform to an energy density. $C_\delta$ and $\psi_0(0)$ are constants determined for the specific wavelet function. In order to determine
the optimum range of frequencies over which to sum in order to reconstruct broad-scale frequency components we generated a time series equal to the sum of three individual sine waves with equal magnitude but specific periods of 1, 0.5 and 0.33 years. This allowed us to quantify the "spread" of spectral information in the wavelet coefficient matrix and define the thresholds to reconstruct the data.

To minimize edge effects associated with the Fourier transform, we add synthetic data to pad the start and end of the time series. For our calculation we repeat the first (last) three years of data backward (forward), accounting for a growth rate based on following (preceding) years. We also 'zero pad' the time series so that the number of points used is an integral power of two, which is necessary as the wavelet transform takes place in Fourier space. The addition of the padded data
allows utilisation of the edges of the time series by ensuring that there is negligible additional error introduced by edge effects, but uncertainty in the spectral decomposition is still likely to be largest at these points. The padded data at the edges of the time series are removed post wavelet decomposition and prior to analysis.

We quantify the numerical error associated with the wavelet transform by applying it to synthetic
time series, which are representative of $CH_4$ time series with a prescribed trend. We find that the value for $C_\delta$ previously reported (Torrence and Compo, 1998) introduces a small trend in the original minus reconstructed residual, and find that $C_\delta = 0.7785$ results in a much smaller, unbiased residual with a typical value of $< 0.5$ ppb for weekly data. Table 2 shows the wavelet parameter values we used in our analysis.

**Appendix C:  Description of TM5 Atmospheric Transport Model**

We use the Transport Model 5 (TM5, (Krol et al., 2005)) to help interpret the trends in Arctic $CH_4$ concentrations, and to quantify the effect of increased Arctic wetland emissions on observed $CH_4$ concentrations at high-latitude monitoring sites.

TM5 is developed and maintained jointly by the Institute for Marine and Atmospheric Research
Utrecht (IMAU, the Netherlands), the Joint Research Centre (JRC, Italy), the Royal Netherlands Meteorological Institute (KNMI, the Netherlands), NOAA and ESRL (Krol et al., 2005).

We run the model run from $1989-2010$ using meteorological fields from the European Centre for Medium-range Weather Forecasts (ECMWF) ERA-Interim reanalysis product. For our calculations we run the model with a horizontal resolution of $4°$ latitude $\times$ $6°$ longitude 34 levels.





Wetlands are the largest natural source of $CH_4$, and occur in regions that are permanently or seasonally water logged, a broad category that includes high-latitude bogs and tropical swamps. We use a wetland flux inventory (Bergamaschi et al., 2005), which is based on an emission distribution (Matthews, 1989) and on an emission model (Kaplan, 2002) that includes the sensitivity to soil moisture, soil temperature and soil carbon. We describe Arctic wetland emissions as being symmetric

across the growing season, peaking in mid-summer and adding up to 25Tg annually. Although recent work (Zona et al., 2016) has shown that cold season emissions from Arctic tundra could equate to more than 50% of annual emissions in the high-latitudes, the peak flux still occurs during summer when wetland emissions are likely to have a larger effect on observed seasonal $CH_4$ concentrations.

      Other emissions include fossil fuel, agriculture and waste (EDGAR 3.2FT2000, (European Com-

mission, 2009)), biomass burning (Global Fire Emissions Database), and atmospheric chemical loss. Atmospheric chemical loss from the reaction with the hydroxyl radical (OH) is the primary mechanism by which $CH_4$ is removed from the atmosphere. This reaction roughly balances the total atmospheric input of $CH_4$ from sources, however small differences lead to trends in the atmospheric abundance of $CH_4$ as have been observed in recent years. Interannual variability of the OH sink is

expected to be small, within ∼2% (Montzka et al., 2011). This is equivalent to ∼10 Tg$CH_4$/yr, the approximate size of inter-annual variability in $CH_4$ emissions (Bruhwiler et al., 2014). Details of chemical loss fields can be found in (Bergamaschi et al., 2005) and consist of a single, repeating seasonal cycle resulting in a $CH_4$ lifetime of approximately 9.5 years.

      We use a prescribed repeating annual cycle of monthly mean 3-D fields of the OH sink, allowing

us to linearize the chemistry so that we can attribute observed variations to individual sources and geographical regions using tagged tracers. We use tracers for emissions from fossil fuel combustion, agriculture, and natural wetlands.

**Appendix D: Analysis and Interpretation of Data and Models**

**Air Sector Analysis of Continuous Data at BRW**

We use hourly averages from the continuous measurements of surface $CH_4$ at Barrow, Alaska (BRW) in an attempt to quantify changes in local summertime $CH_4$ anomalies associated with wetlands (Figure 6).

      We use hourly average measurements of wind speed and wind direction to filter the $CH_4$ data by air sectors (Dlugokencky et al., 1995) (Figure 6). Measurements of $CH_4$ inbound from the North air

sector (20°-110°) are typically used to avoid contamination from 'non-background' or local sources, including gas wells and the town of Barrow and emissions from the large areas of permafrost-affected wetlands in the region. Here we also use measurements of $CH_4$ from the South air sector (135°-220°), which contains information about the local emissions as well as regionally representative air. We use $CH_4$ measurements only when the wind speed is ≥1m/s and ensure that the selected meteo-



370 rological conditions have been in effect for at least one hour. Examination of data from the North and South air sector indicates that there is significant diurnal variability of the $CH_4$ concentration from the south air sector, with a maximum between 0300-0500 local time. To reduce potential biases without over-constraining the data, we calculate diurnal averages from 0900-1700 location, representing the period of least variability. Finally, we determine monthly averages from the filtered data. Figure 7

375 shows that prevailing winds from the North result in substantially more observations from the North air sector than from the South air sector. Air from the South air sector typically has much higher $CH_4$ concentrations as a result of local emissions. The diurnal variability in $CH_4$ is much greater in the air from the south air sector, with the greatest variability in summer. This may be due to $CH_4$ building up in the nocturnal boundary layer and to the recirculation of air during onshore/offshore

380 winds.

 We find that the data collected is better represented by air from the North air sector. To reduce this measurement bias we use a curve-fitting procedure for comparison with the data averages. Figure 8 shows the BRW hourly $CH_4$ observations filtered according to air sector. There are some striking differences between the $CH_4$ concentrations inbound from the north and south air sectors. We show

385 that the seasonal maxima from the south air sector typically occurs during the ascending shoulder of the seasonal cycle and with occasional overlap with the seasonal minima from the North air sector. The strong seasonal differences occur primarily in summer and autumn are consistent with wetland emissions, which typically peak in late summer. Local anthropogenic sources are likely to play a minor role in these differences. We also show the difference between the North and South $CH_4$

390 concentrations are small during the North air sector seasonal maxima, which occurs during winter when local wetland emissions are likely to be negligible.

 We calculate the 'South-North' difference (SND) by subtracting the monthly-averaged $CH_4$ concentrations from the north air sector from those of the south air sector. In theory, this will remove the part of the $CH_4$ concentration that is representative of large, well-mixed air masses, leaving pri-

395 marily the $CH_4$ anomalies resulting from sources local to BRW. We compare these $CH_4$ anomalies with an analysis of BRW 2 m temperature data. We use the number of days $>0°C$ as a temperature metric, representing the potential period of soil thaw and wetland $CH_4$ emission, although emission of $CH_4$ may continue until the soil has frozen over in Autumn (Zona et al., 2016).

 Figure 9 shows that there is generally a positive relationship between $CH_4$ anomalies and the

400 number of summertime days $>°C$ ($5.4\pm9.0$ ppb/decade) but it is not statistically significant (p>0.1). We find that using different approaches to filter the data (not shown) does not affect year-to-year variability of $CH_4$, but it does alter the magnitude (not the sign) of the linear regression coefficient associated with summertime $CH_4$ anomalies at BRW. This a consequence of few $CH_4$ observations from the south air sector that impacts sampling bias. This is as expected since this site was chosen

405 to be representative of background conditions. Consequently, we are confident that there has been



an increase in local summertime CH$_4$ anomalies over the measurement period, but it is difficult to quantify.

Nevertheless, we find that for some periods the temporal variations of CH$_4$ and of surface temperature coincide. The linear trend in the the number of days $>0°$C over the same time span is
8.2$\pm$7.7 days/decade (p$<$0.01), with a particularly rapid increase following 2000. There is a weak relationship between the number of days $>°$C and the CH$_4$ anomalies (r$^2$=0.18), but it is statistically significant, with a sensitivity of 0.47$\pm$0.42 ppb per additional day $>0°$C (p$<$0.05), where some peaks in CH$_4$ (e.g. 2004) do not coincide with peaks in temperature. This relationship remains significant for a range of CH$_4$ filtering criteria which gives us some confidence that at least some
of the observed changes in CH$_4$ could be due to physical mechanisms rather than an artifact of data processing or biases. While it appears that there is no trend in the CH$_4$ anomalies, the uncertainty in the SND is large due to unavoidable sampling bias so that this result is not robust.

**Analysis of Seasonal Amplitudes of CH$_4$ Mole Fractions**

Figure 10 shows our analysis of surface CH$_4$ mole fraction data from individual NOAA/ESRL
CGASN high-latitude monitoring sites (Figure 1). These background flask sites sample air that is considered to be representative of large, well-mixed air parcels. We selected high-latitude sites because they are likely to be the most sensitive to changes in large-scale changes in emissions from Arctic wetlands. We find that the regression coefficients calculated from the seasonal amplitude anomalies of the flask data are predominantly negative. We find that there are just two sites that
exhibit a statistically significant trend in amplitude over the time span of the data: BRW (p$<$0.01) and STM (p$<$0.05).

Figure 11 shows a similar analysis for the continuous observations at BRW by taking weekly averages from the north air sector and applying the same filtering before calculating the equivalent amplitude time series. There are occasionally large differences in the interannual variability of the
flask and continuous amplitude anomalies, with the continuous measurements exhibiting a decline in the CH$_4$ seasonal amplitude that persists for longer than the flask data. Observing a similar large-scale feature in both data sets provides further confidence of the result. Figure 11 also shows that the observed trend is caused main by a decrease in the annual component of the time series.

To examine the effect of filtering thresholds on trend detection in the seasonal cycle, we recon-
struct the seasonal cycle over a range of high and low frequency thresholds. Figure 12 shows that the regressions coefficients have low sensitivity to periods $>$12 months, but vary significantly over the range of high-frequency thresholds where the highest sensitivity is to periods $<$2 months.

**Analysis of MERRA Reanalysis 2-m Temperature Data**

We use gridded time series of temperature from the MERRA reanalysis to estimate the period during
which the mean temperature is higher than 0°C each year. As the dataset is monthly, we use spline



interpolation to acquire a smooth curve from which we estimate the beginning and end of this period. Using this approach we build up a time series for each grid cell and estimate the linear regression coefficients over areas of permafrost-affected wetland, where frozen carbon rich soils are vulnerable to surface warming. We also calculate averaged time series over boreal North American and Eurasian

wetland areas, finding linear regression coefficients of $0.38 \pm 0.18$ ppb/yr (p<0.01) and $0.27 \pm 0.14$ ppb/yr (p<0.01) respectively. Figure 1 shows the map of coefficients and the mean time series for North America and Eurasia.

**TM5 Control Experiment**

Our first use of the TM5 control simulation is to test whether the model seasonal cycle is consistent

with the observed seasonal cycle. The control simulation uses repeating a priori emissions but we expect that the seasonal cycle is comparable in shape and amplitude to the observations. First, we isolate the model and observed seasonal cycle using the wavelet transform (using the same filtering criteria as the flask data analysis) and take the mean seasonal cycle for each site.

Figure 13 shows that the mean model seasonal cycle amplitude is typically smaller than observed

values, with the exception of CBA and SHM. The model and observed timing of the seasonal maxima and minima are similar, although on average the model minima has a lag compared the observed seasonal cycle. This may be due to the presence of 'local' wetland emissions that are sampled by our large-scale model but not observed in the data. The NOAA/ESRL BRW bi-weekly flask samples are collected when the wind is from offshore, which mostly negates the contribution from the

local emissions, whereas the model includes $CH_4$ concentrations from onshore and offshore. This model/data discrepancy is particularly apparent at BRW because the site is close to extensive areas of natural wetlands. Most of the other Arctic sites are more remote, so that there is some delay before they pick up the wetland signal and by this time the air is more well-mixed and representative of a larger area. This difference in sampling at BRW and the subsequent over estimate of the early

summer wetland $CH_4$ anomaly in the model $CH_4$ time series has potentially important implications for our interpretation of the model output shown here. Over the entire timespan of the time series, ICE, STM and ZEP are the sites most poorly represented by the model.

Table 1 summarizes the control calculations associated with the interannual variability and trend in amplitude as a result of changes in atmospheric transport of $CH_4$ to each of the Arctic sites.

**TM5 Experiments Sampled at BRW**

Figure 14 shows the $CH_4$ time series from the TM5 simulations sampled at the BRW site. To quantify the impact of these different emission scenarios on the BRW seasonal cycle, we calculate a time series of seasonal amplitude anomalies following the method we use for the observations. Table 1 show the regression coefficients for the northern high-latitude sites; a graphical depiction of these

data can be found in the main paper. For a repeating seasonal cycle of anthropogenic emissions,




the control simulation shows no long-term change in seasonal cycle amplitude, suggesting that variations in atmospheric transport are unlikely to have caused the observed change in amplitude. As we gradually increase the high-latitude wetland emissions, the seasonal amplitude at BRW begins to decrease, but we find this change is only statistically significant when the annual wetland increase is at 2%/yr. Even this increase in wetland emissions, corresponding to a atmospheric mole fraction change of -3.9 ppb/decade, is not sufficiently large to reconcile the observed decrease in amplitude of -5.8 ppb/decade. The simulations that use the EDGAR anthropogenic emissions correspond to a trend in seasonal amplitude of -4.5ppb/decade ($p<0.05$). If the EDGAR emissions are correct this implies an additional increase in high-latitude wetland emissions of 0.73%/yr is required to reconcile the model with observations. However, recent studies using independent atmospheric measurements of $CH_4$ have identified deficiences in the EDGAR $CH_4$ emission inventory (Bergamaschi et al., 2013; Bruhwiler et al., 2014), implying that our high-latitude wetland emission trend is a conservative estimate.

To further understand the observed changes in the seasonal amplitude of atmospheric $CH_4$ mole fractions we split the contributions from anthropogenic and wetland emissions into long-term (trend) and short-term (seasonal) components by fitting a smooth curve (long-term component) to the seasonal minima of the anthropogenic and wetland signals sampled at BRW. To acquire the seasonal component we use the residual after removing the smooth long-term component.

Figure 15 shows that the maxima of the seasonal component of the anthropogenic signal steadily decreases at BRW, with the largest decrease between $1990-2005$. The maxima of this component typically occurs during December/January and overlaps with the maxima of the BRW $CH_4$ seasonal cycle, resulting in a decrease of the BRW seasonal cycle maxima which carries through to the seasonal amplitude. The anthropogenic signal derived from the EDGAR emissions inventory can be further broken down into contributions from fossil fuel emissions and those from agricultural waste. The change in seasonal amplitude results mainly from fossil fuels.

Figure 15 shows that as high-latitude wetland emissions increase, the maxima of the wetland seasonal component increases due to the larger exchange of $CH_4$. Because this maxima overlaps with the minima of the BRW $CH_4$ seasonal cycle it leads to a decrease of the BRW seasonal cycle amplitude.

**TM5 Experiments Sampled at Other Arctic Sites**

Table 1 show the linear regression coefficients for each emission scenario and each site. We find no statistically significant trends in amplitude except for the BRW 2%/yr case. The control simulation coefficients are not significant, which indicates that atmospheric transport is not driving any persistent, long-term change in the seasonal amplitude. Using the seasonal amplitude anomaly metric, we find that most sites do not exhibit sensitivity to a substantial increase in high-latitude wetland emissions. For the scenarios that use the time-dependent EDGAR emissions only BRW exhibits a




significantly significant trend. The anthropogenic emissions typically resuls in a more negative regression coefficient, indicating that many of the sites are sensitive to changes in emissions but to a lesser extent than BRW. A decrease in contribution from fossil fuel emissions during December and
January results in a dampened maxima and reduced seasonal cycle amplitude.

We use the HYSPLIT Lagrangian dispersion model (Stein et al., 2015; Draxler and Hess, 1998) to investigate further the apparent insensitivity of most NOAA Arctic background sites to changes in emissions from Arctic wetlands (Table 1). The driving meteorology for the HYSPLIT model is from the 0.5° NCEP Global Data Assimilation System model (https://www.ncdc.noaa.gov/data-access/
model-data/model-datasets/global-data-assimilation-system-gdas). We calculate 10-day footprints at individual NOAA Arctic background sites, following Gerbig et al. (2003), for the representative year 2012 at the measurements times at BRW, ZEP, ALT, SHM, and CBA. The observational record at STM ended in 2009 so we did not consider this site in our footprint analysis.

Figure 16 shows using the cumulative footprints June–September that BRW is sensitive to wetland
emissions along the north coast of Alaska and far western Canada. ALT, ZEP and ICE are sensitive to oceans or geographical region without significant wetlands such as the Canadian Arctic archipelago. The two Aleutian Island sites, SHM and CBA, have some sensitivity to far eastern Siberia.

To simulate the contribution of wetland emissions to observed $CH_4$ at sampling sites we convolve the footprints for each site with a distribution of estimated wetland emissions from the CarbonTracker-
$CH_4$ global inversion (Bruhwiler et al., 2014). At BRW, we find a significantly higher frequency of detectable signals from wetlands north of 60°N than at the other two high Arctic sites, ALT and ZEP. BRW also receives more detectable wetland signals than ICE, which often sees transport from the North Atlantic during the warm season.

Figure 17 illustrates how the detectability of a change in amplitude associated with a seasonal
wetland emission source at a particular site changes with the timing of seasonal cycle. A wetland emission source that coincides with the ascending (or descending) edge of the seasonal cycle is "smeared" across time making it less likely to be observed than a source peaking close to the peak of the seasonal cycle, which produces a more localized signal. A similar explanation can be made if there is an atmospheric transport lag between a wetland emission and the atmospheric signal being
sampled at a site far downwind.



**Table 1.** Observed and model seasonal amplitude regression coefficients (ppb/yr) for each Arctic NOAA background site. Asterisks denote values with statistical significance.

| | Time-independent fossil fuel emissions | | | | | Time-dependent fossil fuel emissions | | | |
| --- | --- | --- | --- | --- | --- | --- | --- | --- | --- |
| | Observed | Control | 0.5%/yr | 1%/yr | 2%/yr | Control | 0.5%/yr | 1%/yr | 2%/yr |
| ALT | -0.19±0.37 | 0.01±0.37 | 0.01±0.37 | 0.01±0.37 | 0.03±0.37 | -0.27±0.44 | -0.29±0.43 | -0.29±0.43 | -0.29±0.43 |
| BRW | -0.58±0.38* | -0.03±0.39 | -0.12±0.38 | -0.21±0.38 | -0.39±0.38* | -0.45±0.42* | -0.54±0.42* | -0.63±0.42* | -0.81±0.42* |
| CBA | -0.14±0.45 | -0.08±0.32 | -0.10±0.33 | -0.09±0.34 | -0.03±0.36 | -0.36±0.39 | -0.38±0.40 | -0.38±0.41 | -0.34±0.42 |
| ICE | -0.05±0.51 | 0.04±0.26 | 0.02±0.25 | 0.03±0.26 | 0.10±0.28 | -0.08±0.30 | -0.12±0.29 | -0.12±0.28 | -0.09±0.30 |
| SHM | +0.16±0.39 | 0.06±0.50 | 0.05±0.51 | 0.06±0.53 | 0.15±0.58 | -0.25±0.53 | -0.26±0.54 | -0.26±0.56 | -0.21±0.61 |
| STM | -0.45±0.36* | 0.06±0.32 | 0.12±0.33 | 0.18±0.36 | 0.32±0.43 | -0.23±0.39 | -0.19±0.38 | -0.13±0.39 | -0.01±0.44 |
| ZEP | -0.41±0.46 | 0.09±0.34 | 0.09±0.34 | 0.10±0.35 | 0.21±0.36 | -0.35±0.41 | -0.34±0.40 | -0.33±0.41 | -0.23±0.42 |





**Table 2.** Wavelet parameters

| Parameter | $\delta t$=1/12 | $\delta t$=1/52 |
|---|---|---|
| $\delta j$ | 0.25 | 0.01 |
| $s_0$ | $2\delta t$ | $\delta t$ |
| $C_\delta$ | 0.7785 | 0.7785 |
| $\psi_0$ | $\pi^{-\frac{1}{4}}$ | $\pi^{-\frac{1}{4}}$ |



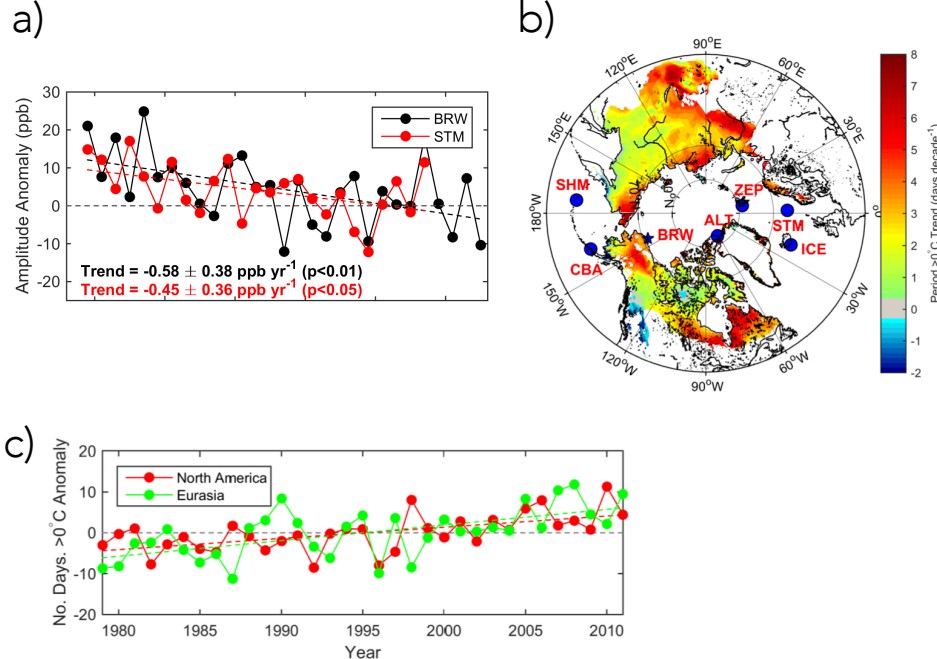

**Figure 1.** a) Amplitude anomaly of $CH_4$ mole fraction measurements (ppb) relative to the long-term time series at Barrow, Alaska (BRW, 156.61°W, 71.32°W) and Ocean Station M, Norway (STM, 66.00°N, 2.00°E). The magnitude and uncertainty of the linear best-fit lines (denoted by broken lines) are shown inset, and the horizontal dashed line denote the line of zero amplitude. b) Decadal trends of potential period of thaw (days >°C) over the polar northern hemisphere during 1979–2012 spatially coincident with permafrost affect wetland areas. Trends are calculated from the NASA MERRA reanalysis dataset. The solid blue circles denote NOAA/ESRL $CH_4$ flask sites used in our analysis, with the blue star over BRW denoting a parallel in situ measurement program. Monitoring sites (not previously defined) include: Alert, Canada (ALT, 62.51°W, 82.45°N), Cold Bay, Alaska (CB, 162.72°W, 55.21°N), Storhofdi, Vestmannaeyjar, Iceland (ICE, 20.29°W, 63.40°N), Shemya Island, Alaska (SHM, 174.13°E, 52.71°N), Ny-Alesund, Svalbard, Norway (ZEP, 11.89°E, 78.91°N). c) Mean annual anomaly of the number of days >°C for North America and Eurasia from 1979–2012.



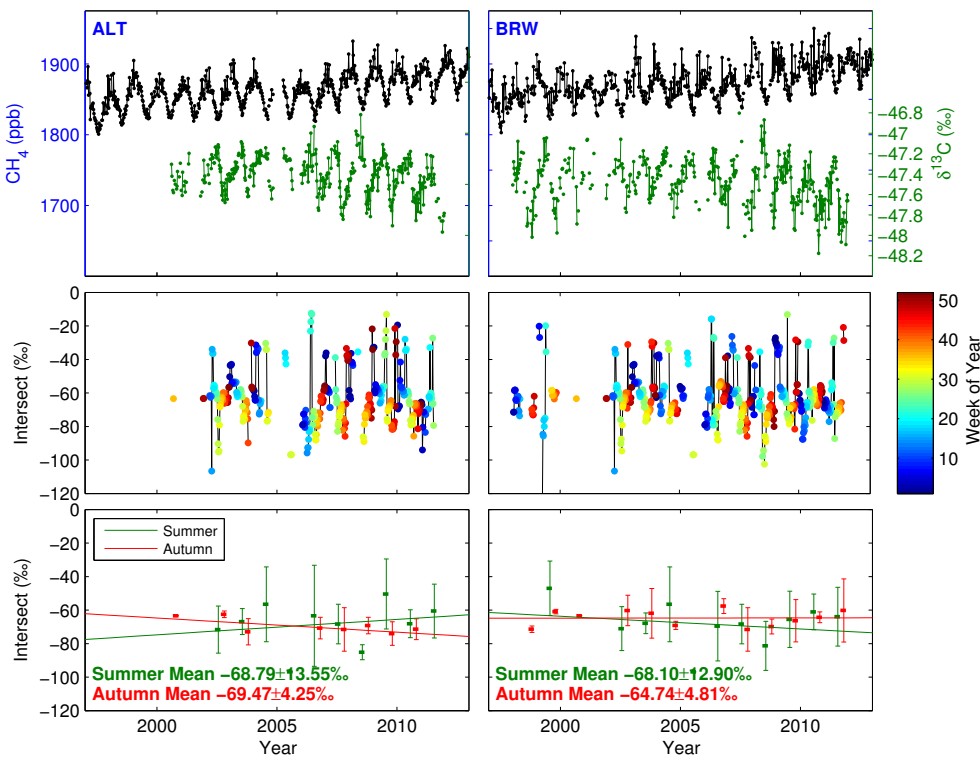

**Figure 2.** (Top panels) $CH_4$ and $\delta^{13}$C-$CH_4$ time series for ALT and BRW, 1997−−2013. (Middle panels) Keeling plot intersects for the 9-week running window calculated from 1/$CH_4$ and $\delta^{13}$C-$CH_4$. Symbol colours denote the week of the year. (Bottom panels) Seasonal mean Keeling plot intersects for Summer (JJA) and Autumn (SON). The error bars represent the standard deviation in each season and the solid lines denote the linear best-fit line of the seasonal means.



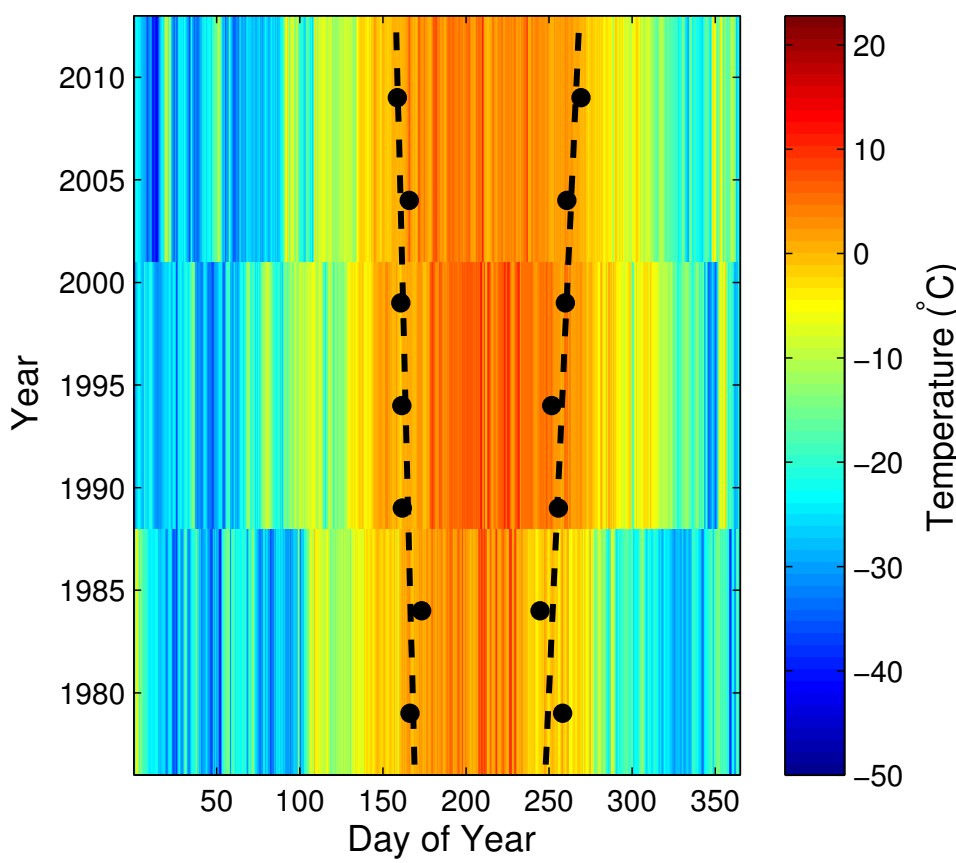

**Figure 3.** Two-metre temperature measurements at BRW, where colour represents changes in temperature seasonally (horizontal axis) and annually (vertical axis). Black markers are 5 year means of the dates when smoothed temperature passes above (left) and below (right) 0°C. Dashed black lines represent the least squares linear fit to the annual zero crossing dates. The time series mean zero crossing dates are approximately day 160 for -ve to +ve and 260 for +ve to -ve respectively.





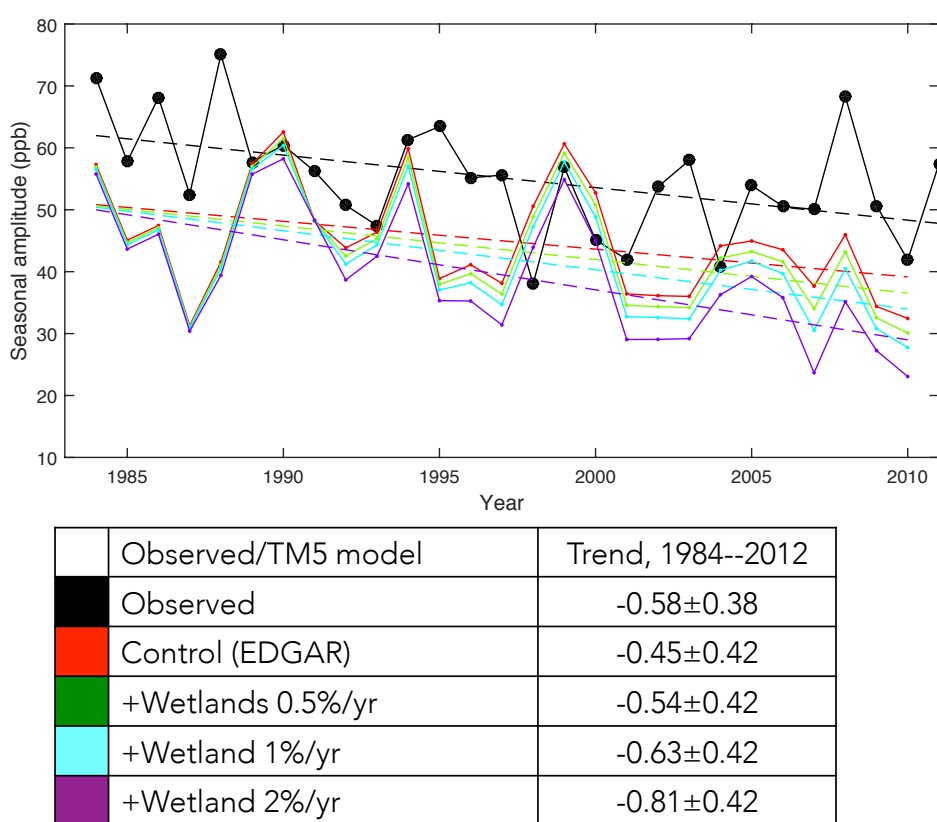

| | Observed/TM5 model | Trend, 1984--2012 |
|---|---|---|
| | Observed | -0.58±0.38 |
| | Control (EDGAR) | -0.45±0.42 |
| | +Wetlands 0.5%/yr | -0.54±0.42 |
| | +Wetland 1%/yr | -0.63±0.42 |
| | +Wetland 2%/yr | -0.81±0.42 |

**Figure 4.** Time series of BRW seasonal cycle amplitude anomalies estimated from the flask data (black) and corresponding values from the TM5 atmospheric chemistry transport model using the EDGAR anthropogenic bottom-up emission inventory and range of boreal wetland emissions scenarios ranging from an annual increase 0.5%/yr to 2%/yr. The dashed lines denote the best-fit lines.





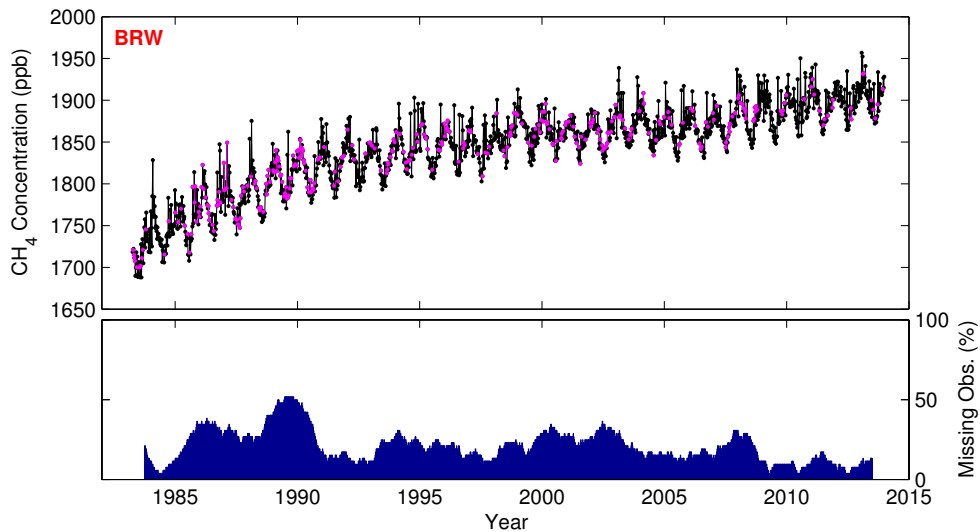

**Figure 5.** (Top panel) BRW CH$_4$ flask-air time series (black), including imputed data (magenta). (Bottom panel) Percentage of missing observations in any one-year period.





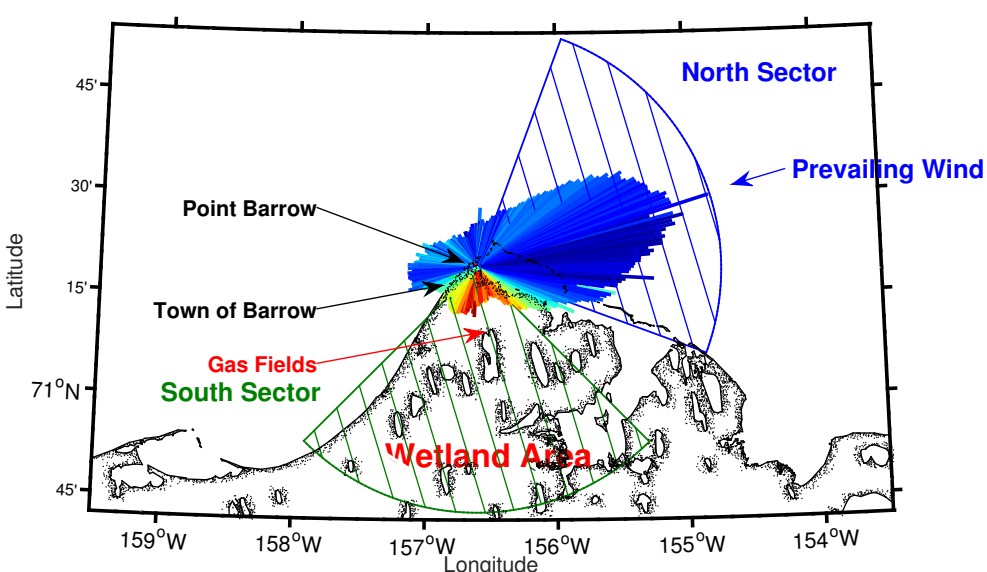

**Figure 6.** Map of the coastal region around BRW. Point Barrow is located at the centre of the wind rose. The wind rose shows the prevailing winds from the north east and is coloured by $CH_4$ concentration, indicating higher concentrations from the south where there are gas fields and wetlands.





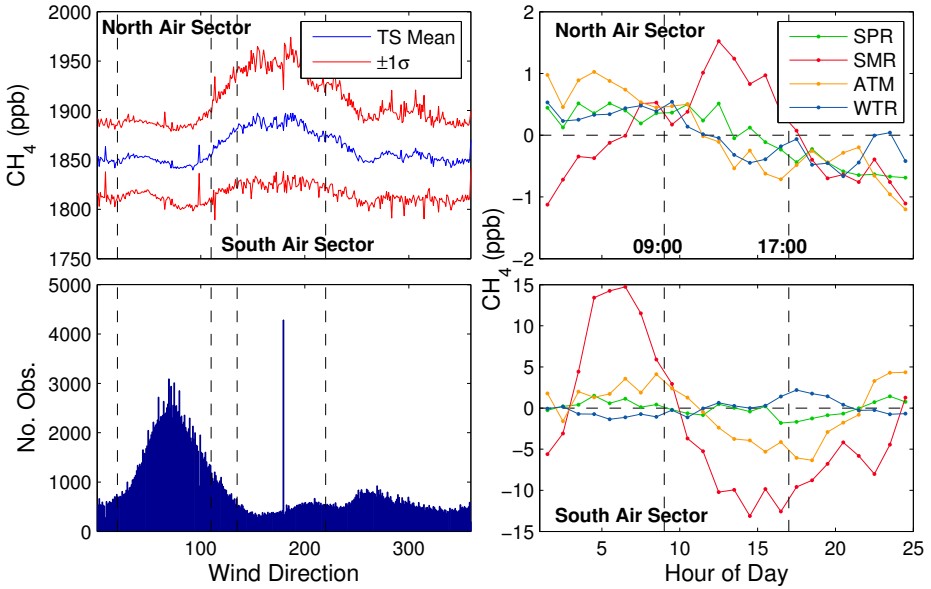

**Figure 7.** (Top left) Mean $CH_4$ concentrations $\pm 1\sigma$ from different wind directions. (Bottom left) Corresponding histogram showing the number of hourly observations influenced from different wind directions. (Top right) Mean seasonal diurnal cycles of $CH_4$ concentrations from the North air sector. (Bottom right) Mean seasonal diurnal cycles of $CH_4$ concentrations from the South air sector.





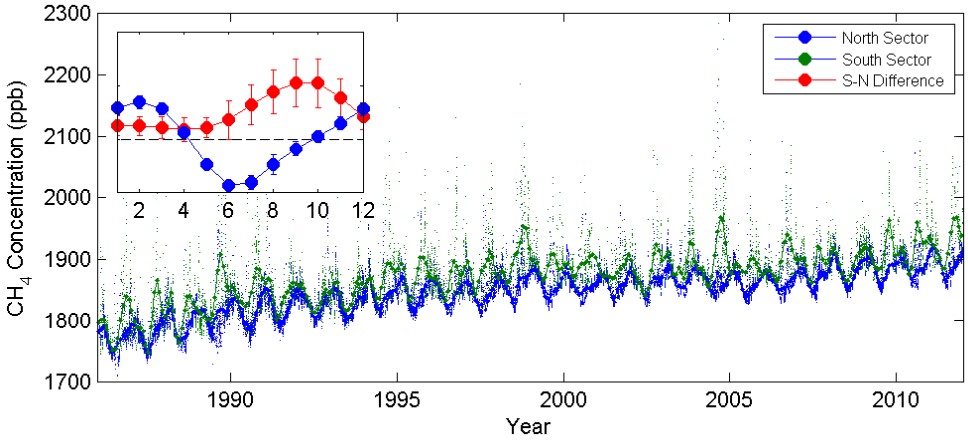

**Figure 8.** CH$_4$ time series filtered according to the criteria defined for the North (blue) and South (green) air sectors. The inset panel shows the mean north sector seasonal cycle, and the mean South-North Difference.





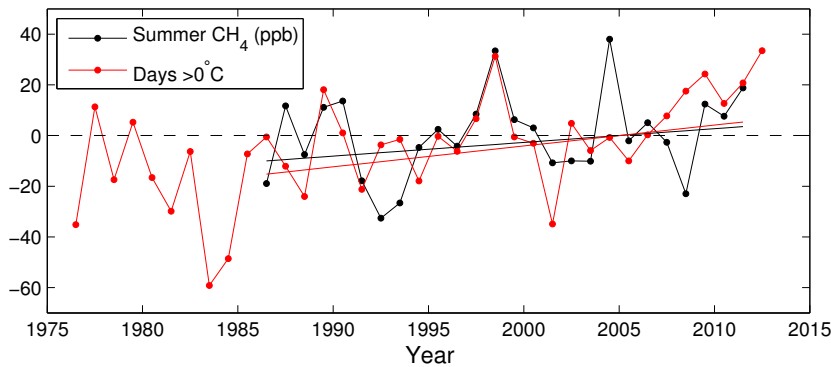

**Figure 9.** Summer CH$_4$ anomaly determined from the continuous data (black) and number of days >0$^\circ$C (red) with linear least squares fit.



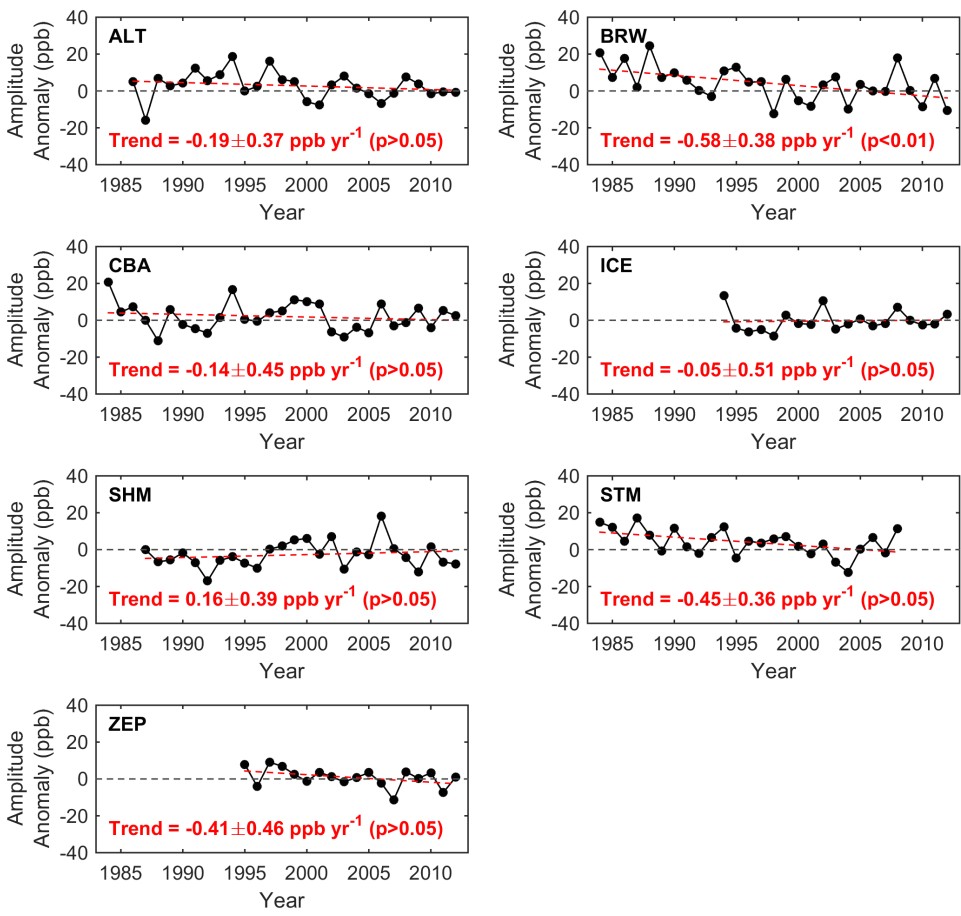

**Figure 10.** Seasonal cycle amplitude anomalies at several high northern latitude sites: Barrow, Alaska (BRW, 156.61°W, 71.32°W) and Ocean Station M, Norway (STM, 66.00°N, 2.00°E), Alert, Canada (ALT, 62.51°W, 82.45 °N), Cold Bay, Alaska (CB, 162.72°W, 55.21°N), Storhofdi, Vestmannaeyjar, Iceland (ICE, 20.29°W, 63.40°N), Shemya Island, Alaska (SHM, 174.13°E, 52.71°N), Ny-Alesund, Svalbard, Norway (ZEP, 11.89°E, 78.91°N). The red dashed line denotes the linear best-fit trend line with the regression coefficients and associated p-value shown inset. The black dashed line denotes the zero amplitude anomaly line.





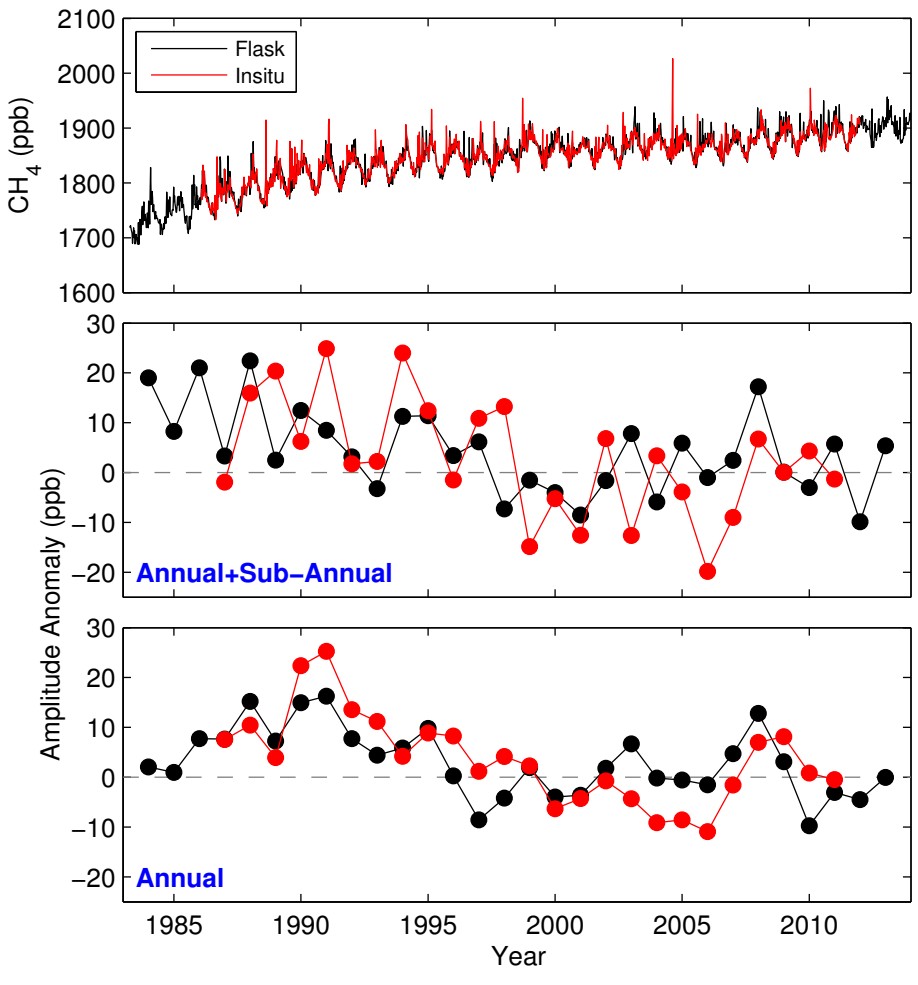

**Figure 11.** (Top panel) Comparison of flask and continuous CH$_4$ concentration. (Middle panel) Amplitude anomalies calculated from the sum of annual and sub-annual frequency components. (Bottom panel) Amplitude anomalies calculated from the annual frequency component.





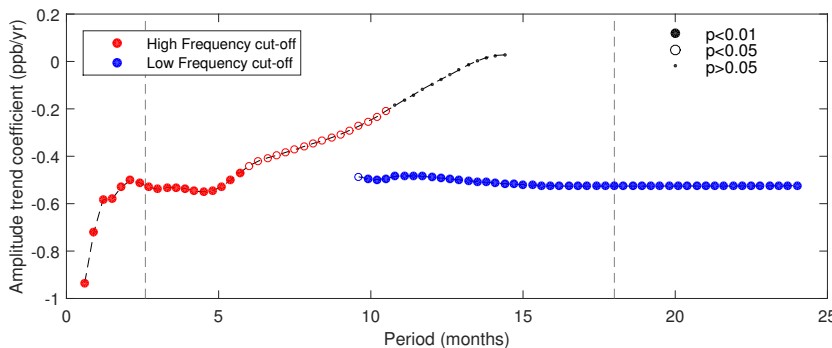

**Figure 12.** Amplitude trend coefficients calculated for a range of high (red) and low (blue) frequency filtering thresholds when reconstructing the CH$_4$ seasonal cycle. Solid and open circles denote coefficients that are statistically significant at the $p<0.01$ and $p<0.05$ levels, respectively. The black dashed lines denote coefficients that are not significant. The two grey, vertical dashed lines show the high and low frequency filtering thresholds chosen for our analysis.

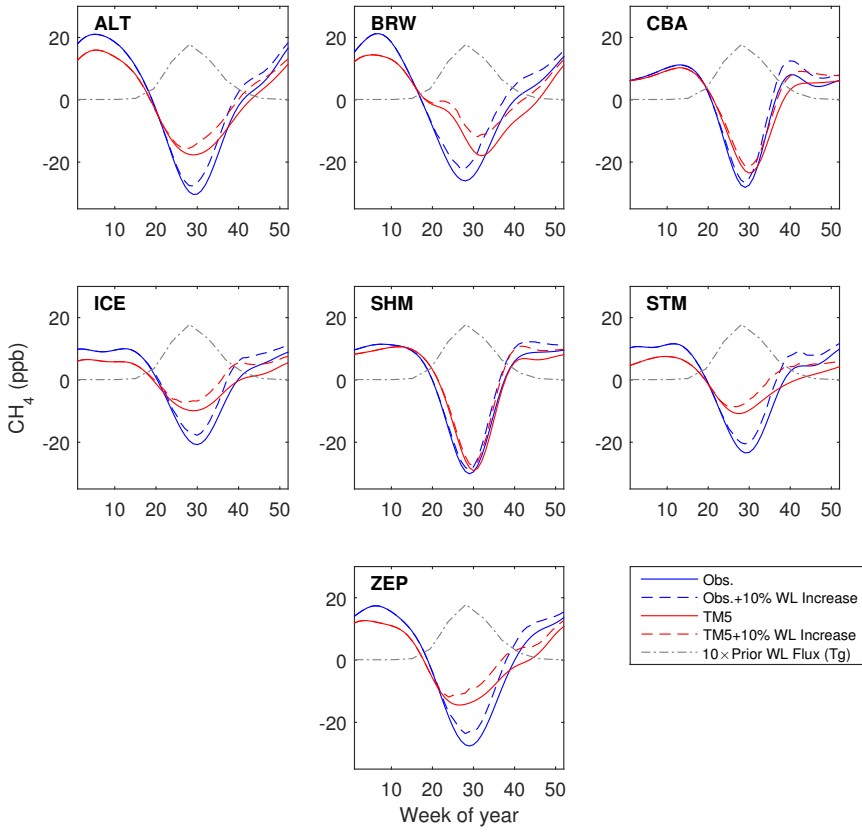

**Figure 13.** Mean observed (blue) and model (red) CH$_4$ seasonal cycles at each high-latitude site (see Figure 10). Dashed lines denote the effect of a 10% increase in wetland emissions. The grey dashed line denote the a priori wetland flux (multiplied by 10 for visual comparison.)





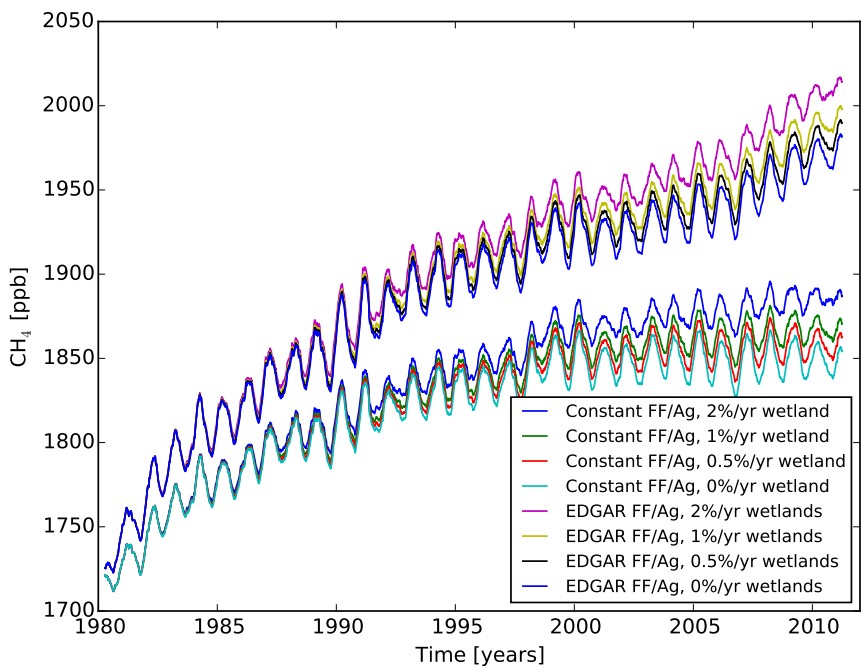

**Figure 14.** Model CH$_4$ time series sampled at the location of BRW from an ensemble of TM5 CH$_4$ simulations described in section C.



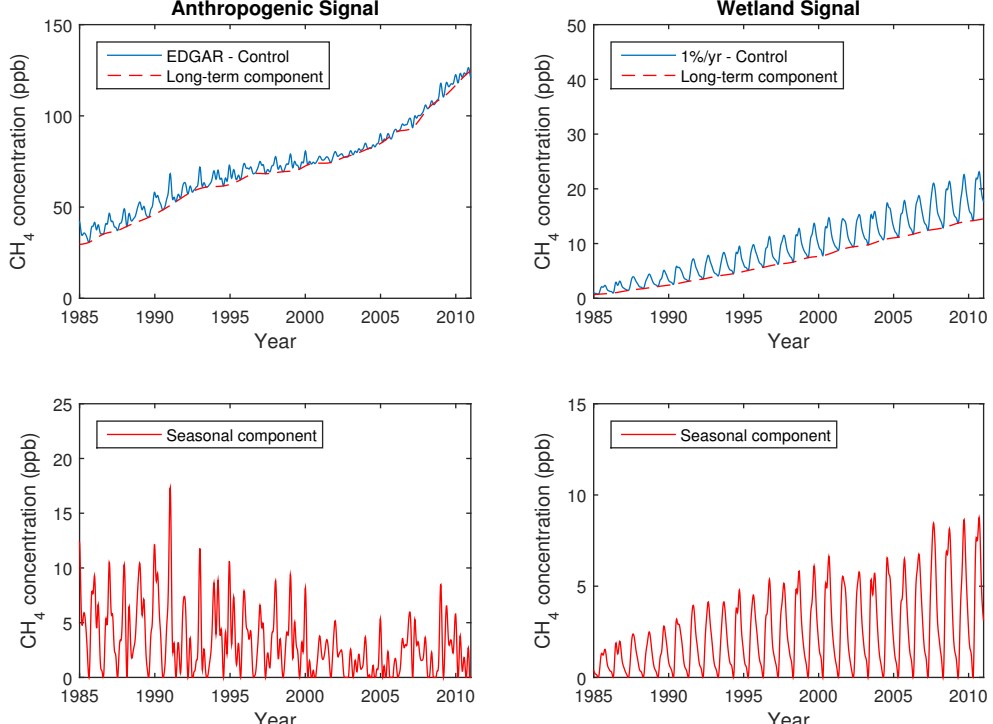

**Figure 15.** (Top panels) Time series of contributions from anthropogenic and wetland emissions to observed BRW $CH_4$ mole fractions. The dashed line is fitted to the minima of each annual cycle and represents the long-term variation of the time series. (Bottom panels) The seasonal contribution of the anthropogenic and wetland contributions to observed BRW $CH_4$ mole fraction, determined by subtracting the long-term component from the $CH_4$ time series.





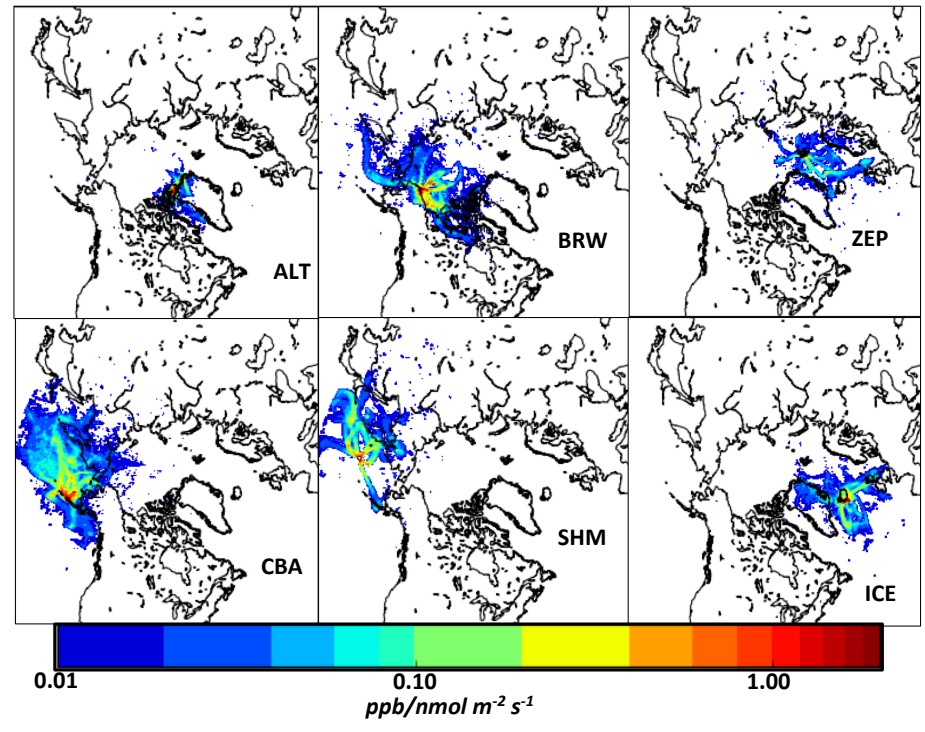

**Figure 16.** Surface sensitivity (ppb/nmol/m$^2$/s) of NOAA Arctic air sampling sites (Figure 1) to Arctic wetland CH$_4$ emissions, calculated using the HYSPLIT dispersion model, June–September 2012.



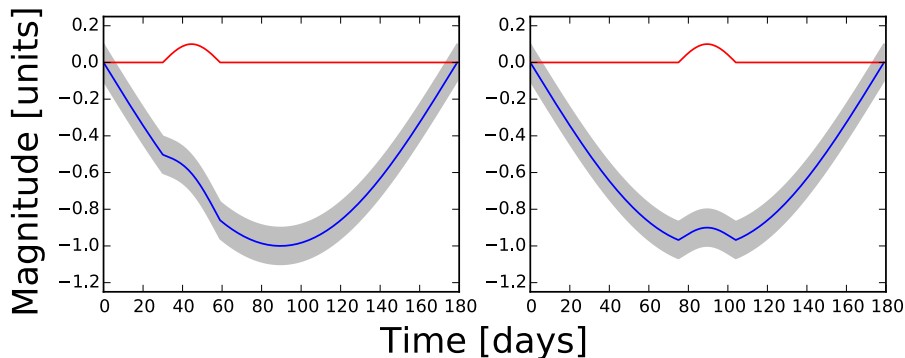

**Figure 17.** The detectability of a 30-day wetland emissions source of atmospheric $CH_4$ denoted by a red solid on a larger observed seasonal cycle of atmospheric $CH_4$ mole fraction denoted by the solid blue line. The grey envelope denotes a 10% uncertainty of the mole fraction data.



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
