# Peer review of "Increasing boreal wetland emissions inferred from reductions in atmospheric CH4 seasonal cycle"

_Atmospheric Chemistry and Physics, 2016_

## Referee Comment (RC1) · Anonymous Referee #1 · 23 Sep 2016

The authors use a wavelet transform to analyze the change in the seasonal cycle of CH4 mole fractions at high latitude sites. They find a reduction in the peak-to-peak amplitude that could be driven by (1) Enhanced wetland emissions in the warm season, counteracting the yearly drawdown by OH in summer (2) Reduction in the CH4 emissions from fossil fuel sources, leading to less wintertime accumulation in the NH high latitudes. They specifically focus on option (1) and try to find evidence using various methods.

Although the wavelet transform seems an interesting approach, the authors fail to demonstrate its advantage over other possible methods (see point 1 below). More-over, the paper is very hard to read, mainly because the main text is separated from

the Appendix (see point 3 below). Many figures are not well introduced nor discussed properly in the text. Also, the scope is very broad, and many data-manipulation methods are employed to demonstrate increasing wetland emissions. Methane isotopes are analyzed, a detailed wind-sector analysis for BRW is performed, and Lagrangian footprint models are invoked. After a long and tedious analysis of the paper, I am however not convinced (point 3 below) and have the impression that the authors developed a strong preference for increasing Arctic emissions. The model experiments seem to indicate that most Arctic stations show small sensitivity for Arctic wetland emissions, with the exception of BRW. It is also clear, however, that the skill of the model in reproducing the seasonal cycle is rather poor (figure 13), which casts additional doubts on the predictive value of the sensitivity experiments. I think the paper needs a serious rewrite and should also be substantially reduced in scope and size. However, more attention should be given to the advantages of the wavelet analysis over other methods.

Below, I summarize four main points of criticism, after which I list some minor comments.

1 The use of wavelet analysis

In principle, may methods could be used to derive the peak-to-peak amplitude in methane mixing ratios using atmospheric observations. The authors claim that they "improve on the Fourier transform". However, it remains unclear why the most logical analysis from simple monthly mean mixing ratios would not be suitable for the analysis. The answer may be trivial (e.g. low frequency noise needs to be filtered out, e.g. line 60), but this step is vital if you want to sell a rather complicated analysis tool to filter your data. In that sense, the paper makes an unbalanced impression, leaving out vital information like this, and presenting rather long separate analysis in the Appendices. NOAA provides software to filter time-series (used in their data visualization tools) and I wonder what is wrong with that? So Appendix B, and maybe the main text, should better motivate the use of Wavelet transforms.

[Figure]

**2 The set-up of the numerical experiments**

The authors present two sets of simulations with "tagged" methane, meaning they separate wetland methane from fossil methane. One set contains a repeated year of EDGAR anthropogenic emissions, and one set time-dependent emissions. The authors furthermore perturb Arctic emissions by 0.5, 1 and 2% per year. I find this set-up strange. First, the system is basically linear, because I believe the feed-back of methane on OH is not included. So, you can freely combine "CH4-arctic wetlands", CH4-anthropogenic", and "CH4-tropical wetlands", to arrive at total CH4 in any given source mix. Furthermore, they sample the model at NOAA/ESRL monitoring sites as diurnal averages, which are averaged further to weekly time resolution. This is also a step back from the commonly applied co-sampling at measurement stations. I understand that a homogeneous time-series is needed for the wavelet transform. However, this could still result in some bias, because the measurements are gap-filled in a different way than the model (e.g. recognized on line 460). I think these issue do not invalidate the results of this study per-se, but I get the impression that the simulations are not well-thought-thru. Anyhow, the way the model represents the seasonal cycle at some high-latitude stations is worrisome (figure 13).

**3 Appendices & Structure**

By using many Appendices, the paper gets messy. For instance, Appendix D part 1 (air sector analysis) seems a paper in itself and the added value of all this extra material for the evidence seems limited. From where is appears in the main text, I would expect that the BRW analysis would look closely at 13CH4, but this surprisingly is not done. Also, I found myself switching between main text and appendices, because the main text is by far not stand alone. For instance, the "Interpretation of numerical experiments" reads verypoorly. First, increasing wetland emissions seem to work ONLY for BRW. In contrast, fossil emission reductions seem to decrease the seasonal amplitude at all stations and explain 75% of the observed amplitude reduction at BRW. Line 152: consequently, a smaller coincident increase . . ..in wetland emissions of 0.73% per

year is needed. Accounting for biases in EDGAR, this number could be larger. This only becomes clear after reading the Appendix. Moreover, it seems the authors are desperate to proof an increase in high-latitude wetland emissions. My interpretation of the above facts would be: reductions in fossil emissions lead to a reduction in the seasonal amplitude of CH4 mole fractions (but do not exclude an increase in wetland emissions). Most clearly this is read on line 166: "we could not quantitatively reconcile observed and model trends in the amplitude of the seasonal cycle. Only by including . . . atmospheric CH4". As far as I can see, this was also the only modification of the methane emissions that the authors explored (they also looked at two versions of anthropogenic emissions), so this result is not particularly surprising. Also, the abstract mentions only the 0.7%/yr increase in wetland emissions, even while most high-latitude stations appear not sensitive to wetland emissions. In conclusions, this firm conclusion seems an overstatement.

4 Comparison to other studies

Although the approach is original by focusing on the peak-to-peak amplitude of the seasonal cycle, the authors should still compare their results to other studies. Formal inverse modelling studies have been published including the high-latitude stations to constrain methane sources. These studies thus implicitly account for changes in the peak-to-peak amplitudes, by looking for a least-squares fit with all observations. Bergamaschi et al. (Bergamaschi, P. et al. Atmospheric CH4 in the first decade of the 21st century: Inverse modeling analysis using SCIAMACHY satellite retrievals and NOAA surface measurements. J Geophys Res 118, 7350–7369 (2013).), however, found no increase in arctic wetland emissions. At least some discussion of the findings of existing studies should be included.

Minor issues:

Line 104: Figure 1a misses x-axis

Line 111: Not true for the Montzka paper. This only talks about inter-annual variations

in OH, and not about long-term changes.

Line 113: If the relatively heavy fossil CH4 emissions would decline, this would imply overall lighter emissions also, so I do not see why this argument is not mentioned here.

Line 117: "Each Keeling plot". Sounds a bit unexplained to me. How are these Keeling plots made? From monthly data? Only from co-sampled CH4 and 13CH4? I see that the caption mentions 9-week running window. But that seems strange, given the fact that gaps seem different for CH4 and 13C-CH4.

Line 141: -0.45 +/- 42 ppb/year ???? typo?

Line 163: hydroxyl radical? I did not see what was presented on this issue, at least not in the main text.

Line 183: The authors suggest experiments in which the seasonal cycle of the emissions is modified. I am a bit surprised that these simulations are not part of the current paper. It seems a rather interesting and logical complementing way to further explore the potential impact of Arctic emissions.

Line 350: 2% is too small. The correct number is 2.3 +/- 1.5%

Line 353: 9,5 years should be 9.4 years. Unclear where the OH comes from (Spivakovsky, 2000 or calculated with TM5 chemistry?).

Line 374: Figure 7 is not well explained. Apparently SPR, SMR, ATM, and WTR stand for the seasons. The link from the text to the figures is once again very weak.

Line 405: "Consequently . . .quantify". This statement is non-scientific and should be removed.

Line 419: Tell also here what "our analysis" means.

Line 433: main mainly

Line 486: It would help to describe the nature of the EDGAR deficiencies, and how this

leads to a conservative trend in arctic wetland emissions.

Line 534: I do not really see what this adds. The only reasonable experiment I could think of is to widen the seasonal wetland emissions like in figure 3 and to investigate its impact on the seasonal cycle of simulated methane at the arctic stations.

Figure 13: What does "Obs + 10% WL increase" mean?

---

## Referee Comment (RC2) · Anonymous Referee #2 · 4 Oct 2016

This study investigates trends in the seasonal cycle amplitude in long-term CH4 records from high-latitude measurement sites to investigate trends in emissions from natural wetlands. The analysis leads to the conclusion that high latitude wetland emissions must have increased by at least 0.7%/yr. This in itself is a very relevant and significant finding. However, as will be explained below, it remains unclear how this number is derived and what it really means. In addition, the description is difficult to follow as at jumps back and forth between different topics. Overall, significant revisions will be needed to make this study acceptable for publication in ACP.

GENERAL COMMENTS

To use the observed seasonal cycle amplitude in northern latitudes to investigate
changes in wetland emissions sounds like a logical idea, but it should first be tested if it works. The authors do the experiment needed for that, but don't draw any conclusion from the answer. Table 1 shows the change in seasonal amplitude when natural emissions are increased in the model. The assumption is that seasonal cycle amplitudes reduce as wetland emissions increase, but the model shows this only for BRW. This is the only station where the observed trend is significant and the relationship holds. Indeed, looking at the numbers, I conclude that the trend of 0.7%/yr is based only in this site. The corresponding emission trend of 5Tg/30 year is an extrapolation of this percentage to the whole boreal-arctic. To derive such an estimate from just a single site is brave, but this is true even more so if the approach is essentially falsified at the 6 other sites that are analyzed. It could even be worse: The authors analyze the impact of sampling by wind sector. It remains unclear, however, if the trend in Table 1 is only for the clean marine air sector or also for the locally influenced continental sector. In the latter case the extrapolation to the whole boreal-arctic would certainly be invalid. Just to use the un-flagged flask data from Barrow would not be sufficient I'm afraid. Interestingly the authors notice that the seasonal amplitude is not very sensitive to wetland emissions. In that case, why is it used as a metric in the first place? Wouldn't it have been better to use the asymmetry of the shoulder seasons or something like that? Looking at Figure 13, the emission does maximize near the seasonal cycle minimum. However, the signal tends to increase with the integral of the emissions, i.e. as summertime emissions fill the arctic reservoir. Due to the atmospheric mix of these emissions you expect that the integrated signal becomes more representative of the whole arctic as the season progresses. Indeed the impact of wetlands is more similar across the sites near week 40, than week 30. It indicates that the seasonal minimum may make the method overly sensitive to regional influences.

The TM5 model has the nice characteristic that its transport is linear. Therefore you only need a single run with increasing wetland emissions to calculate the impact of any trend that is a multiple of that trend. Looking at Table 1 one can easily check if this is the case. It is for BRW, but not for any of the other sites. Something must have gone

wrong.

Many plots are presented, several of which don't seem needed to support the conclusions, but could be made available as supplementary material. Actually, there seems to be confusion between appendices and supplementary information. On the other hand, if all information in the appendices would move to supplementary material then the main text would not be self-contained anymore. Some significant restructuring is needed to solve this issue. When prioritizing figures, a new one is needed to demonstrate the performance of the wavelet method. Other tools have been used in the past. Moreover, the claim is made that the wavelet method is better than the Fourier transform method, but none of this is demonstrated. It should be shown how well the method works, especially given the need for some substantial data padding and gap filling.

SPECIFIC COMMENTS

page 13, line 433: How can a trend in seasonal amplitude be most clear in the annual component of a time series?

page 14, line 445: With is MERRA T2m regressed against?

page 14, line 460: The model could also be sampled for offshore conditions for, example, by choosing 1 grid box further into sea.

Table 1: Some table notes are needed to explain the numbers in the various columns without having to search the mean text for explanation (plus minus represents what?).

Figure 2, middle panels: The most visual jump in color is between the end and the beginning of the year, whereas those weeks are close together in time. The color bar should be made 'circular'.

Figure 2: The purpose of this figure is not really clear. We know that the signature of d13C is depleted in this range. It would be more relevant to see any trend in seasonality. Since this information can probably not be obtained from the data, I wonder what it is that we learn here.

Figure 3: What causes the vertical line structure in this figure? It seems like every day in the year is smoothed with the same day in other years. What is done and why?

Figure 5: How can imputed data seem to be out of the range of adjacent measured data?

Figure 9: What is on the y-axes here? Don't you need two y-axes for CH4 and number of days?

Figure 12: What combinations of low and high cut offs are used? (I mean, if one is varied, then what is the other?)

TECHNICAL CORRECTIONS

page 4, line 126: 'get been getting'

page 5, line 132: 'Figure 1b' i.o. 'Figure 1'

page 10, line 332: 'run the model run'

page 11, line 373: '09:00-17:00 location'

page 13, line 433: 'mainly' i.o. 'main'

Figure 1a: missing labels on the x-axis

Figure 7: Explain the legend in the right panels.

Figure 17: What is 'Magnitude (Unit)'?

―――――――――――――――――